# Pediatric Ventilation Skills by Non-Healthcare Students: Effectiveness, Self-Perception, and Preference

**DOI:** 10.3390/ijerph20043026

**Published:** 2023-02-09

**Authors:** Santiago Martínez-Isasi, Cristina Jorge-Soto, Cecilia Castro-Fernández, Clara Baltar-Lorenzo, María Sobrido-Prieto, Jose Manteiga-Urbón, Roberto Barcala-Furelos

**Affiliations:** 1CLINURSID Research Group, University of Santiago de Compostela, 15782 Santiago de Compostela, Spain; 2SICRUS Research Group, Institute of Health Research of Santiago de Compostela (IDIS), 15706 Santiago de Compostela, Spain; 3Nursing Faculty of Santiago de Compostela, University of Santiago de Compostela, 15782 Santiago de Compostela, Spain; 4Health Sciences Department, University of A Coruña, 15071 A Coruña, Spain; 5Pediatric Emergencies Unit, Biomedic Research Institute of A Coruña—INIBIC, University Hospital of A Coruña (CHUAC), SERGAS, 15006 A Coruña, Spain; 6REMOSS Research Group, Faculty of Education and Sport Sciences, University of Vigo, 36005 Pontevedra, Spain

**Keywords:** simulation training, learning, bystanders, emergency care, ventilations, pediatric basic life support

## Abstract

Since a great number of infant cardiopulmonary arrests occur outside of the hospital, it is crucial to train laypersons in cardiopulmonary resuscitation techniques, especially those professionals that will work with infants and children. The main objective of this study was to evaluate the efectiveness of ventilations performed by professional training students. The secondary objective was to analyze the preference between different ventilation and chest-compression methods. The sample consisted of 32 professional training students, 15 preschool students, and 17 physical education students. The activity was conducted separately for each group, and we provided a 10 min theoretical training about infant basic life support followed by a 45 min practical training using a Laerdal Little Anne QCPR CPR manikin. A practical test in pairs was organized to record the ventilation as performed by the participants, establishing the difference between the efficacious and the non-efficacious ones. Furthermore, we handed out a survey before and after training to evaluate their knowledge. More than 90% of the students completely agreed with the importance of learning cardiopulmonary resuscitation techniques for their professional future. More than half of the sample considered that they perform the rescue breathings with the mouth-to-mouth method better. We observed that through mouth-to-mouth-nose ventilations, the number of effective ventilations was significantly higher than the effective ventilations provided by a self-inflating bag and mask (EffectiveMtoMN 6.42 ± 4.27 vs. EffectiveMask 4.75 ± 3.63 (*p* = 0.007)), which was the preferred method. In terms of the compression method, hands encircling the chest was preferred by more than 85% of students. Mouth-to-mouth nose ventilation is more efficient than bag-face-mask ventilation in CPR as performed by professional training and physical activity students. This fact must be considered to provide higher-quality training sessions to professional training students.

## 1. Introduction

Early bystander cardiopulmonary resuscitation (CPR) is essential for survival when an out-of-hospital cardiac arrest (OHCA) occurs. In fact, although more than half of OHCA are witnessed, only 46% of these witnesses perform CPR maneuvers, according to a study carried out in the USA in which 3900 child cardiorespiratory arrests (CRA) were analyzed [1]. In Europe, according to the results of the Eureka Two survey, witnesses initiate resuscitation maneuvers in 58% of cases, but the variability between countries is very high [2].

Fortunately, the estimated incidence of OHCA in children is low, ranging from 8 to 20 cases per 100,000 children per year [3]. However, as a CRA is a time-dependent event, providing early care has an important impact on survival and the development of aftermath.

CRA in children is usually caused by respiratory phenomena such as suffocation, choking, or a foreign-body airway obstruction unlike adult CRA, which usually has a cardiac source. Therefore, ventilation is especially important in pediatric cases, which is why current guidelines recommend combining ventilation and compression [4]. When analyzing the survival rates in unwitnessed or respiratory arrests, these are much lower in children who only received compressions (8% survival at 30 days) compared to those resuscitated with compressions and ventilations (13%) [5].

When performing compressions on infants, there are several techniques: with both thumbs over the center of the chest and surrounding the entire chest with the rest of the hands and compressing with both thumbs or with the index and middle fingers. The first technique is recommended when there is only one rescuer, while the second technique is more applicable when there are two rescuers [4].

Recently, a new compression method for two rescuers was studied: the Smereka method, which proposes using the two thumbs facing each other without hugging the chest, folding the rest of the fingers of the hands against the palms. This method could provide a more adequate compression depth and a better compression–ventilation ratio than the traditional one [6].

As different methods to carry out the CPR sequence exist, it is important that professionals who work with children previously train in these skills in order to know their disposition and determine the method that makes them feel most comfortable. This is why the aim of this study was to evaluate the effectiveness of rescue breathings and compressions performed by students from a non-health field while determining the basic life support (BLS) methods with which they felt most comfortable.

## 2. Material and Methods

Study design

A quasi-experimental study was carried out, in which a brief theoretical and practical training was given to the 32 participants involved.

This study included the participation of a convenience sample formed by professional training (PT), including physical activity and sports technician students and early-childhood-education technician students from a concerted educational institution located in Santiago de Compostela in the north of Spain, which took place in January 2020. 

In order to carry out the participant selection, the following inclusion criteria were considered: age equal to or over 18 years, being enrolled at the time of the research in the selected center, and studying a non-health field mention. As exclusion criteria, we defined the following: having completed training in life support during the current academic year and having a physical and/or mental disability that prevented them from carrying out the knowledge assessment.

Intervention

In order to accomplish the training activity, the participants were divided according to their current studies, obtaining two groups: the physical activity and sport group and the early-childhood-education group.

The training consisted of a brief, 10 min theoretical presentation of pediatric CPR, which included survival chain explanations, ventilation methods (mouth-to-mouth and self-inflating bag mask), and the instruction in two compression methods to apply in cases where two rescuers are available (traditional and Smereka).

Thereafter, students participated in a 45 min practical training. For the practical training, the participants were divided into small groups, with a participant/instructor ratio of 6/1. The participants rotated through a circuit of simulation stations; each one was dedicated to training a specific skill:First scenario: chest compressions using the two-thumb encircling hands technique and mouth-to-mouth-nose ventilation performed by two rescuers;Second scenario: chest compressions using two thumbs facing each other (Smereka method) and mouth-to-mouth-nose ventilation performed by two rescuers;Third scenario: only ventilation with a mask and self-inflating bag.

After the training, the instructed skills were evaluated using a pediatric CRA simulated scenario attended in a first intervention by two rescuers. The resuscitation maneuvers were carried out for 4 min, with a role change between the rescuers after two minutes. Throughout the whole training, the participants received quality feedback from the instructors.

To execute the chest compressions, each participant was allowed to choose the compression method with which they believed that they compressed more efficiently. The options provided were the Smereka method [6] (Figure 1) and the two-thumb encircling hands technique [7] (Figure 2), the same that were provided during practical training.

Mouth-to-mouth ventilation and ventilation with a mask and self-inflating bag skills were evaluated in different PCR simulation scenarios.

During the practical test, the ventilations performed were recorded through an evaluation sheet, identifying those that had been effective. Only the breathings that visibly elevated the manikin’s thorax were considered effective in the study.

Each participant’s performance was recorded on video with a mobile phone to be able to review the algorithm if necessary.

A knowledge assessment was carried out using a questionnaire before and immediately after the training. 

Variables:

We recorded the following demographic variables for each participant: sex (male/female) and age (in years).

Regarding the resuscitation skills, the following were recorded: total ventilations performed with mouth-to-mouth-nose ventilation (TotalMtoMN), effective mouth-to-mouth-nose ventilations (EffectiveMtoMN), total ventilations administered with masks and self-inflating bag (TotalMask), effective ventilations with mask and self-inflating bag (EffectiveMask), first choice of compression method variable (Smereka or hands encircling chest), and second choice of compression method variable (Smereka or hands encircling chest). The self-perceived quality of ventilation was recorded in the post-training knowledge assessment using a visual analogue scale (Figure 3) as given below:

Materials:

A Laerdal Little Anne QCPR CPR manikin was used for simulation because it provided feedback throughout the process. CPR quality standards were set in accordance with the recommendations of the guidelines [4].

Regarding the surveys, the prior knowledge questionnaire consisted of 18 questions divided into four blocks: 1st block: prior training (3 questions);2nd block: general knowledge about resuscitation (6 questions);3rd block: knowledge about pediatric CPR (5 questions);4th block: opinion (4 questions).

The post-training knowledge questionnaire consisted of 15 questions (blocks 2, 3, and 4 of the previous questionnaire) and a question on the self-perceived quality of ventilation.

All variables were recorded on the data sheet for each participant.

Ethics:

Participation in this study was voluntary, and the participants could withdraw at any time without repercussions, nor was any incentive granted to favor it. Initially, the design and objective of the study were both explained to the participants, emphasizing the implications of their participation in it, and an informed consent was given, in which the information confidentiality provided was guaranteed according to the Organic Law 3/2018 of 5 December 5 regarding the subject of Personal Data Protection. Furthermore, the researchers’ contact information was indicated to provide a proper communication.

This study complies with the principles of the Declaration of Helsinki and has obtained a favorable report from the local Ethical Review Board (Faculty of Science Education and Physical Activity, University of Vigo, Spain).

Statistical analyses:

Qualitatives are indicated with absolute frequency and percentage and quantitatives with mean and standard deviation.

The Kolmogorov–Smirnov test was performed to analyze the homogeneity of the quantitative variables, and the Levene test was carried out to analyze the homogeneity of the variances. For the comparisons between groups, the chi-square test and Mc-Nemar-Bowker Test were used for qualitative variables, and the Student’s *t*-test or Mann–Whitney U tests were used for quantitative variables depending on normal distribution. 

The statistical package SPSS for Mac, version 25, was used to enable the statistical analysis, and a significance level of *p* < 0.005 was established in all comparisons.

## 3. Results

The study sample consisted of 32 professional training students, with 17 of them being physical activity and sports technician students (53.12%) and 15 being early-childhood-education technician students (46.8%). The demographic characteristics of the sample are detailed in Table 1.

Before the training session, more than 90% of participants considered that CPR knowledge was important in the context of their future profession. Likewise, 27 of 32 (84.37%) considered that a specific subject regarding CPR training is necessary and that it should be compulsory to obtain an education.

Twenty participants (62.5%) had received previous CPR training, half of them while they were enrolled in previous official studies. The average duration of this previously received training was 1.5 h. Despite this, less than 35% of them considered that they were capable of performing CPR, and only nine participants (28%) perceived their skill as effective or sufficient.

Concerning CPR knowledge, twenty-nine participants (90.62%) knew the emergency telephone number before the training, but this percentage reached 100% after it. Remarkably, none of the CPR quality parameters evaluated (rhythm, depth, and compression-ventilation ratio) were known by more than 37% of the sample, with average depth being the variable that achieved the worst results before training.

Knowledge of adult CPR before and after training is detailed in the following table (Table 2).

Regarding pediatric CPR, seven participants (22%) considered that they knew how to perform it before the training. After the training, this question was answered positively by 100% of the sample.

Only one participant knew the quality parameters before the training. However, after the session, more than 40% of the participants answered each of these questions correctly.

In addition to the quality parameters, the existence of multiple chest-compression methods in infants was also unknown to the sample.

Corresponding to breathings evaluation, all the participants performed a breath cycle with mouth-to-mouth nose method and another one with a mask and self-inflating bag. Three hundred and ten breaths were provided using the mouth-to-mouth nose method (values for all participants), of which 199 (64.2%) were efficacious, which were understood as those that raised the manikin’s thorax. With a self-inflating bag and face mask, 318 ventilations were performed, of which 142 (44.65%) were efficacious.

We also observed that through mouth-to-mouth nose ventilations, the number of effective ventilations was significantly higher than the effective ventilations provided by a self-inflating bag and mask (EffectiveMtoMN 6.42 ± 4.27 vs. EffectiveMask 4.75 ± 3.63 (*p* = 0.007)). The ventilations recording is shown in Figure 4.

When comparing the effective ventilations according to the grouping variable “sex”, it was shown that women performed more effective ventilations with both the mouth-to-mouth method (woman 8.56 ± 2.874 vs. man 4.13 ± 4.406 (*p* = 0.009)) and the mask-and-bag method (woman 6.38 ± 2.473 vs. man 3.13 ± 3.931 (*p* = 0.016)).

In reference to the self-perception of the ventilation ability, 18 (56.3%) participants considered that they ventilated better with the mouth-to-mouth method, 10 (31.3%) with a mask and self-inflating bag, and only 4 (12.5%) did not identify differences in their proficiency with both methods.

Additionally, more than 85% of the participants decided to perform compressions using the hug method both in the first and in the second simulation, and this method was chosen over the Smereka method.

## 4. Discussion

The present study analyzes the skills that a selected group of vocational students had for infant ventilation as well as their general knowledge of pediatric and adult CPR. In this study, it was observed that knowledge of CPR quality parameters was very low in relation to adults and null in relation to children. On the other hand, the percentage of effective ventilations was higher in mouth-to-mouth ventilations. In addition, this study captures the participants’ preferences for both compression and ventilation methods. There is currently no other published study that has carried out these records outside the health professional field. 

The preference in both ventilations and compression methods must be taken into consideration by life support trainers in order to adapt the training program to the population to which it is directed. If a lay rescuer with little or no training and experience feels insecure about the algorithm maneuvers, the quality of CPR will presumably drop, and in many cases, these feelings may cause CPR to be delayed or omitted, as reflected by the varying rates of bystander-initiated CPR in Europe [1,2,8]. According to the evidence, witnesses initiate resuscitation maneuvers in 58% of cases of an out-of-hospital cardiac arrest in Europe, but the variability between countries is very wide (between 13–82.6%) [2], and in most cases, bystanders provide compression-only CPR.

In the context of a pediatric CRA, ventilations are of greater importance since asphyxia is usually the cause [4]. As the available evidence shows, ventilation is the worst-performed skill during CPR both in students [9,10] and in healthcare professionals [11,12], even causing negative consequences to the victim [13,14].

In our study, the percentage of effective ventilations was significantly higher with the mouth-to-mouth nose method. Similar results were obtained by Santos Folgar [15] in a study carried out with 46 nursing students evaluated through a 4 min simulated scenario with a role change after 2 min; those students received a 4 h theoretical and practical training. In the study, the percentage of ventilations with adequate volume was significantly higher with the mouth-to-mouth nose method; in addition, a higher percentage of ventilations with excessive volumes was recorded with a bag and mask. These results agree with the few studies carried out, such as that of Madden in 2006 [10], which indicated that ventilated volume was the worst skill of nursing students during adult CPR.

The worse performance of bag-mask ventilations in both health sciences students [16] and healthcare professionals [17] can be attributed to the difficulty of managing instrumental manual ventilation in infants. If this is the case, training sessions in the management of instrumental manual ventilation in infants and children should be reinforced in all groups.

More than half of our sample indicated that they provided better ventilations through the mouth-to-mouth nose method. This self-perceived quality correlates with the effectiveness of the breaths provided, which was higher in mouth-to-mouth nose breaths when compared to breaths delivered through bag and mask. Participants in the Santos-Folgar study [15] also reported better self-assessment quality of mouth-to-nose ventilations than with a bag and mask although the perception of quality was low with both methods.

An important aspect of practical resuscitation training is the correct self-perception of abilities since it has been observed that this does not always correspond to reality, as abilities are often overestimated [18,19]. On the other hand, in studies carried out with students, these abilities were often underestimated, which may be related to age, as they are in a formative period and have minimal professional experience.

It should be noted that this study is the first to record the effectiveness of self-perceived ventilation in a non-healthcare population. In health professionals, the relationship between resuscitation skills and self-perceived quality has been studied since a sufficient level of knowledge and skill in CPR is expected of them. Therefore, it is important that both students and professionals can be considered appropriately qualified. However, in the case of an OHCA, the first responder in most cases is not a healthcare worker but a layperson, and the available evidence indicates that the population’s skills in BLS are scarce [20]. In addition to this fact, despite witnessing a CRA, witnesses do not always initiate resuscitation maneuvers. Therefore, knowing their own abilities could improve the percentage of bystanders who initiate resuscitation maneuvers in the case of witnessing an OHCA and at the same time be a motivation for laypeople to perform a greater number of retraining sessions to maintain and improve their abilities.

One of the problems with CPR ventilations is that, in some of the attempts, it is not possible to insufflate air into the lungs either due to a bad head-neck position or due to failures in the seal between the mouth or mask and the child’s face. In our study, participants were able to insufflate air in more than 64% of mouth-to-nose ventilations and almost 45% of bag-mask ventilations. These results are lower than those obtained by nursing students [15] and by lifeguards [21], where percentages close to 100% were observed in mouth-to-mouth and around 75% in mask and bag. However, our participants received shorter training than in the cited studies and did not have prior health training, unlike other populations studied such as nursing students or lifeguards.

Our results show that professional training students ventilated better with mouth-to-nose during the infant CRA simulation. This is consistent with published evidence where mouth-to-nose ventilation has been shown to be the most effective skill. Practice in simulation models and clinical experience can be expected to improve bag-mask ventilation skills, as has been observed in studies conducted with healthcare personnel [22].

To summarize, the low performance found in the results of this study is possibly due to the lack of prior training of our participants. This fact supports the initiative to include first aid content in all levels of training, which should be a prioritized line of research in the future.

However, the results obtained by our participants do not differ much from those obtained in studies carried out with health professionals or students of health forces, which leads us to consider the hypothesis that frequent retraining is just as important as initial training [9,10,11,15].

Regarding this study’s limitations, it should be taken into consideration that manikin use is common in the training and evaluation field of life support skills, but they do not accurately reflect reality. This fact limits the extrapolation of the obtained data, as occurs in all simulation studies.

On the other hand, participants were in a training activity context, and this affects behavioral aspects related to motivation or stress, which are very different from those of a real CRA context, which affects the validity of the results. Furthermore, knowledge and skills were evaluated immediately after training; thus, we could not assess the impact of this very brief training program in the middle or long term.

This study was carried out with a convenience sample; all the participants belonged to the same geographical, social, and cultural context, which may affect the representativeness of the sample, but as already indicated, there is no study of similar characteristics carried out in a field other than the health sector. Lastly, breathings were simply assessed as effective or not based on whether they allowed for obvious expansion of the manikin’s chest, but the insufflated volume was not quantified, so excessive volume breaths were not identified. However, given the context in which it was carried out, “provide ventilation that allows air entry” is the only feedback that a first responder has in the event of an OHCA.

## 5. Conclusions

Mouth-to-mouth nose ventilation is more efficient than bag-face mask ventilation in CPR performed by professional training and physical activity students with an infant-simulated model. This fact must be taken into account in order to provide higher-quality training sessions to PT students.

In addition, students consider that it is crucial to receive specific training in CPR during their formative period.

## Figures and Tables

**Figure 1 ijerph-20-03026-f001:**
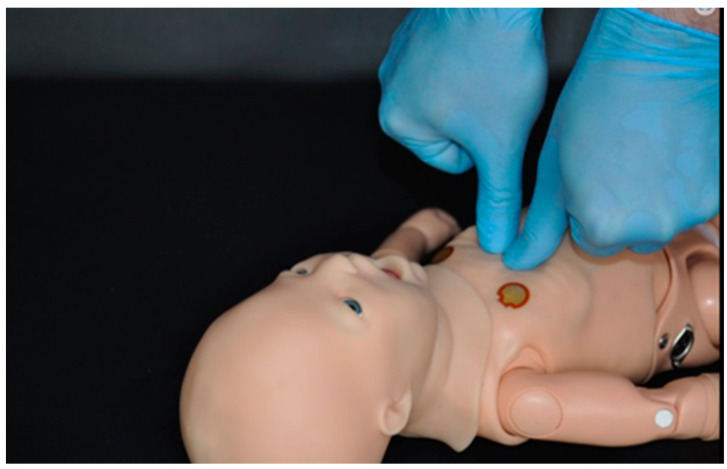
Smereka chest-compression method [6].

**Figure 2 ijerph-20-03026-f002:**
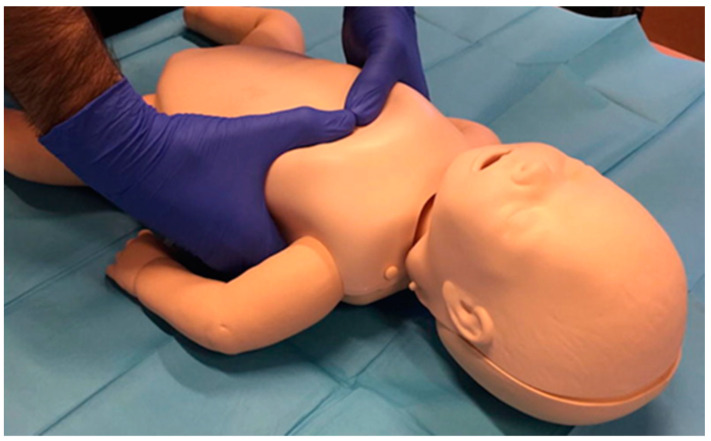
Newborn chest compression by encircling chest with hands [7].

**Figure 3 ijerph-20-03026-f003:**
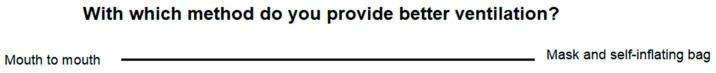
Visual analogue scale to register the self-perceived quality of ventilation included in the survey. Own production.

**Figure 4 ijerph-20-03026-f004:**
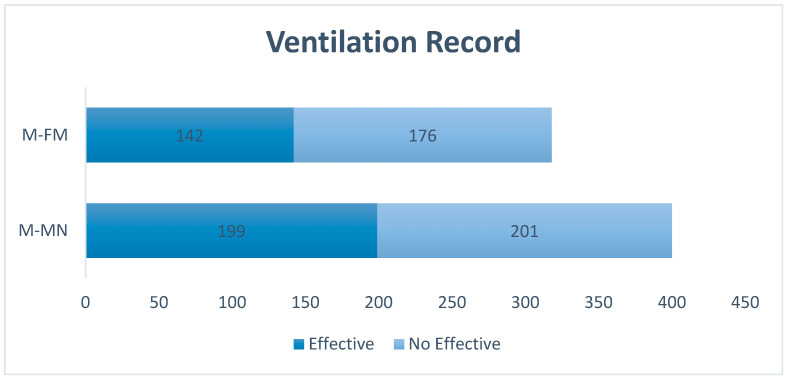
Ventilations record performed during simulation. Data presented in absolute frequency. M-MF Total ventilations with self-inflating bag and face mask. M-MN total ventilations by mouth-to-mouth nose.

**Table 1 ijerph-20-03026-t001:** Sample demographic characteristics. Data are presented as absolute frequency (percentage) or average (standard deviation).

Variables	Early Childhood Education	Physical Activity and Sports	*p*-Value
Participants	15 (46.88%)	17 (53.2%)	
Sex Men/Women	2 (13.3%)/13 (86.7%)	14 (82.4%)/3 (17.6%)	<0.001 ^a^
Age (mean ± SD)	18.87 ± 4984	19.65 ± 1869	0.682 ^b^

^a^ chi-square test. ^b^ Student’s *t*-test.

**Table 2 ijerph-20-03026-t002:** Pre- and post-training knowledge assessment. Data shown as absolute frequency (percentage).

Variable	PRE Training	Immediately Post Training	*p*-Value ^a^
CPR Notion	25 (78.12)	32 (100)	- ^b^
Do you know how to perform an adult CPR?	11 (34.37)	22 (68.75)	0.01
CPR skill self-perceived as efficient or enough	9 (28.12)	15 (46.87)	<0.001
Correct compression:ventilation ratio	12 (37.5)	23 (71.87)	0.001
Correct chest compression	2 (6.25)	13 (40.62)	0.003
Correct chest compression depth	3 (9.37)	16 (50)	- ^b^
Pediatric CPR Knowledge			
Do you know how to perform a newborn CPR?	7 (21.87)	32 (100)	1
Correct compression:ventilation ratio	1 (3.12)	25 (78.12)	<0.001
Knowledge of 3 different methods of newborn chest compression	1 (3.12)	19 (59.37)	<0.001
Correct chest compression rate	0	14 (43.75)	- ^b^
Correct chest compression depth	1 (3.12)	17 (53.12)	- ^b^

^a^ McNemar–Bowker Test ^b^ No statistics have been calculated because the variable is a constant.

## Data Availability

Not applicable.

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
