# Peer review of "Pediatric Ventilation Skills by Non-Healthcare Students: Effectiveness, Self-Perception, and Preference"

_ijerph, 2023, doi:10.3390/ijerph20043026_

Round 1

Reviewer 1 Report

Congratulations to the authors. 

Some minor changes:

1. Aim:

-   The aim of the study was to evaluate the effectiveness of ventilations performed by Professional Training students. However, the authors carried out the study in a simulated scenario. For this reason, it would be more appropriated to use the concept of ‘efficacy’.

2. Introduction

-      Great

3. Methods

-      Line 80: *inclusion criteria*.

-   Could the author please include the sample size in ‘material and methods’ section?

-    Could the authors please include the description of the assessed variables before the report of measuring instruments?

-  The authors describe the variables which were analyzed in the knowledge questionnaire in the ‘materials’ section. Could the authors please include these variables in the ’variables’ section?

-      Line 180: * Statistical analyses*

-      Line 181: duplicated (line 163).

4. Results

-      Table 1: *chi-square test*.

-   Table 1. Could the authors please include *p-value* instead of *p*?

5. Discussion

-    Great

6. Conclusions

-    Great

Author Response

In this file, we address the comments and suggestions made by the reviewers, to whom we are grateful as we consider that their contributions will substantially improve this manuscript.

Reviewer 2 Report

In their manuscript the authors have tested the knowledge and performance of 32 students (of non-health related studies) for infant ventilation.

An interesting study of importance. I enjoyed reading it. However, a number of corrections and clarifications should be performed.

Here are my specific comments:

·         Please format the manuscript according to the journal´s requirements. For instance, remove the words “Introduction”, “Aim” etc from the abstract. And do not use the dots in the methods section before the subheadings “study design”, “intervention” etc.

·         Line 21: Does this mean that the 10min theory and the 45 min practice training was performed separately for all 32 students?

·         Line 25: Define the abbreviation “CPR” on first mention.

·         Abstract in general: not upper case letters in “Professional Training”, “Preschool” and “Physical Education” (and in the rest of the manuscript”

·         Line 26: “considered that they performed rescue breathings better with the mouth-to-mouth…”

·         Line 27: “We observed that…”

·         Line 28: “was significantly higher…”

·         Line 29: Are the values means plus/minus what? And please write plus/minus instead of “+” only.

·         Line 30: What do you mean by “traditional one”?

·         Line 33: The information that an “infant simulated model” was used should be presented earlier in the abstract; in the methods part.

·         Line 45: “high” instead of “wide”

·         Line 52:”…in pediatric cases”

·         Line 58: “hands”

·         Line 71: Define “BLS”

·         Line 80: “inclusion”

·         Line 81: Remove “old”

·         Line 82: “As exclusion criteria we defined…”

·         Lines 90-93: In the abstract one would think that this we performed separately, it is not mentioned here. Please clarify.

·         Line 94: “Thereafter, students participated in a 45….”

·         The sentence in line 98 should be moved to line 105.

·         Line 97: “a specific skill:”

·         The sentence in line 109-111 is complicated and can be shortened to something like “…each participant was allowed to chose the compression method with which they believed that they compressed more efficiently.”

·         No references to Figure 1 and 2 can be found in the text.

·         The line numbering 113-128 wet wrong (with the two Figures). Please correct.

·         Line 129: Were these two additional scenarios? This in unclear.

·         Line 136: Which questionnaire? This could be presented as a Supplementary File.

·         Lines 143ff: Please write 1st, 2nd etc.

·         Line 157: No reference to Figure 3 can be found in the text.

·         Line 181 can be removed. It is redundant.

·         Lin 186: t test or Mann-Whitney U Test and not “and”. “were used depending on normal distribution”. The Chi squared test is not mentioned in this section.

·         Lines 186 and 189: Use “comparisons” instead of “contrasts”

·         Line 193: “and 15” instead of “the 15 left”

·         Line 196: “Data are presented as absolute frequency (percentage) or mean (standard deviation)”

·         Table 1: The percentage of participants does not add up to 100.  Chi squared instead of “cuadrado”. Mean instead of average. “Student T test”. “.” Instead of “,” in the age line.

·         Line 200: “27 of 32 considered…”

·         Line 201: “education”

·         Line 214: “Data shown as absolute frequency…”

·         Table 2: What do you mean by “CPR notion”?

·         Table 2: A statistical test comparing the results pre- and post-training should be performed (Chi squared test)

·         Table 2: A “)” is missing following 21.78

·         Table 2: What do you mean by “-“? 0?

·         Line 216: Should be 22%

·         In line 226 and 227 are these the values of all participants taken together? As mean? Please clarify.

·         Line 229: “We also observed that…”

·         Line 230: “was significantly higher”

·         The values in line 231 are these means and SD? Please clarify.

·         Lines 232 and 234: This is Figure 4.

·         Line 235: I do not understand the abbreviation M-MF and in the Figure M-FM is written.

·         Line 238: “…women performed more effective…”

·         Line 239 and 240: are the values mean and SD? Please clarify.

·         Line 239: “.” Instead of “,”

·         Line 242: “considered”

·         Line 243: “did”

·         Line 245: “Additionally” instead of “going further”

·         Line 246: “was chosen”

·         In the first paragraph of the discussion the main findings should be shown.

·         Line 263: “are of greater importance” instead of “acquire greater relevance”

Author Response

(The authors gave the same response as above.)

Reviewer 3 Report

Review of the manuscript “Pediatric Ventilation skills by non-healthcare students: effectiveness and self-perception”

The study is a quasi-experimental study aiming to evaluate the effectiveness of ventilations performed by Professional Training students and, as a secondary objective to analyze the preference between different ventilation and chest compression methods.

Title

The authors cite the information about participants´ preferences on compression and ventilation methods as a strength of the study, therefore it should be in the title as well. Suggestion: Pediatric Ventilation by non-healthcare students: effectiveness, self-perception, and preferences.  

Introduction

1.      The introduction is well written and has an adequate length.  

Methods

2.      Study design should cite the fact that Professional Training students could be either Physical Activity and Sports Technician students or Early Childhood Education Technician students.

3.      Did all participants receive both theoretical and practical training by the same instructors?

4.      Was the ventilation performance evaluated by the same person for each participant?

Results

5.      Was there any participant excluded?

Discussion

1.      Authors should discuss more about the low performance found in their results and create hypothesis to improve the performance of non-health professionals trained in CPR.

2.      Also, should include as a limitation or discuss the fact that performance was measured immediate after training, and it does not measure knowledge retention.  

Conclusion

3.      The conclusion is appropriated and it is in accordance with the results found.

Author Response

(The authors gave the same response as above.)

Round 2

Reviewer 2 Report

The authors have performed the requested changes. The manuscript has improved a lot. Some minor things remain:

- Abstract, line 25: I would stay with "effective" (also rest of the Abstract and manuscript)

- Abstract: I apologize for the misunderstanding with the plus/minus. I meant the symbol "±". Please use the symbol (also in the rest of the manuscript).

- The McNemar-Bowker Test (Table 2) is not described in the Statistical Methods section.

- line 141: CPR?

- Line 180 and 181: remove "variable"

. Line 235: "considered"

- Line 236: I would leave "degree" in the text.

- Line 367: After the statement ".in studies carried out with health professionals or students of health forces," references are needed.

- Line 377: "we couldn´t assessed" change to "we could not assess"